# Molecular Characterization of Ovarian Yolk Sac Tumor (OYST)

**DOI:** 10.3390/cancers13020220

**Published:** 2021-01-09

**Authors:** Khalil Hodroj, Aleksandra Stevovic, Valery Attignon, Domenico Ferraioli, Pierre Meeus, Sabrina Croce, Nicolas Chopin, Lea Rossi, Anne Floquet, Christine Rousset-Jablonski, Olivier Tredan, Frédéric Guyon, Isabelle Treilleux, Corinne Rannou, Marie Morfouace, Isabelle Ray-Coquard

**Affiliations:** 1Centre Léon Berard (CLB), 69008 Lyon, France; Valery.Attignon@lyon.unicancer.fr (V.A.); Domenico.Ferraioli@lyon.unicancer.fr (D.F.); pierre.meeus@lyon.unicancer.fr (P.M.); nicolas.chopin@lyon.unicancer.fr (N.C.); lea.rossi@lyon.unicancer.fr (L.R.); christine.rousset-jablonski@lyon.unicancer.fr (C.R.-J.); olivier.tredan@lyon.unicancer.fr (O.T.); isabelle.treilleux@lyon.unicancer.fr (I.T.); corinne.rannou@lyon.unicancer.fr (C.R.); isabelle.ray-coquard@lyon.unicancer.fr (I.R.-C.); 2EORTC, Translational Research, 1200 Brussels, Belgium; aleksandra.stevovic@eortc.org (A.S.); marie.morfouace@eortc.org (M.M.); 3Institut Bergonié, 33000 Bordeaux, France; s.croce@bordeaux.unicancer.fr (S.C.); a.floquet@bordeaux.unicancer.fr (A.F.); F.Guyon@bordeaux.unicancer.fr (F.G.)

**Keywords:** OYST, molecular characteristics, targetable mutation, patient outcome

## Abstract

**Simple Summary:**

Ovarian yolk sac tumors (OYSTs) are rare and specific therapeutic strategies are needed after the failure of platinum-based first-line and salvage regimens. This retrospective study included ten patients with OYST, including patients with relapsed disease and disease-free patients. Three patients (33.3%) harbored oncogenic mutations in *KRAS*, *KIT* and *ARID1A,* which may be used as a target. Our series shows that relapsed patients with molecular analysis had clinically relevant molecular alterations. Future research with dedicated trials and multicenter international collaborations are needed to demonstrate the efficacy of specific therapeutic strategies after failure of platinum-based first-line and salvage regimens.

**Abstract:**

Most patients with malignant ovarian germ cell tumors (MOGTCs) have a very good prognosis and chemotherapy provides curative treatment; however, patients with yolk sac tumors (OYSTs) have a significantly worse prognosis. OYSTs are rare tumors and promising results are expected with the use of specific therapeutic strategies after the failure of platinum-based first-line and salvage regimens. We initiated a project in collaboration with EORTC SPECTA, to explore the molecular characteristics of OYSTs. The pilot project used retrospective samples from ten OYST relapsed and disease-free patients. Each patient had a molecular analysis performed with FoundationOne CDx describing the following variables according to the Foundation Medicine Incorporation (FMI): alteration type (SNV, deletion), actionable gene alteration, therapies approved in EU (for patient’s tumor type and other tumor types), tumor mutational burden (TMB), and microsatellite instability (MSI) status. A total of 10 patients with OYST diagnosed between 2007 and 2017 had a molecular analysis. A molecular alteration was identified in four patients (40%). A subset of three patients (33.3% of all patients) harbored targetable oncogenic mutations in *KRAS*, *KIT*, *ARID1A*. Two patients at relapse harbored a targetable mutation. This retrospective study identifies clinically relevant molecular alterations for all relapsed patients with molecular analysis. Dedicated studies are needed to demonstrate the efficacy of specific therapeutic strategies after the failure of platinum-based first-line and salvage regimens and to explore the potential relationship of a molecular alteration and patient outcome.

## 1. Introduction

Malignant germ cell tumors (MGCTs) represent 5% of all ovarian cancers and 80% of the pre-adolescent malignant ovarian tumors. Most patients with ovarian germ cell tumors will be cured with first-line therapy, with five-year overall survival rates of 95.6% and 73.2% in stage I and advanced stages, respectively [1].

The malignant ovarian GCTs (mOGCTs) are assumed to derive from primordial germ cells (PGCs) with inherited or somatically acquired alterations [2]. PGCs may be identified in human embryos at 5–6 weeks of gestational age. Orchestrated by the KIT ligand (KITLG, also known as the stem cell factor, SCF) and its receptor KIT, and the chemokine SDF1 (CXCL12) and its receptor CXCR4, PGCs migrate from the proximal epiblast (yolk sac) through the hindgut and mesentery to the genital ridge and become gonocytes [3,4]. Level of gene expression and appropriate timing of expression are crucial to appropriately control the process.

The primitive GCTs are subdivided into the ovarian counterpart of the male testicular seminoma, dysgerminoma (DG), and non-DGs. The development of non-DGs is characterized by differentiation into cell histologies that mimic embryonic tissues (embryonal carcinoma (EC), teratoma) and extraembryonic tissues (yolk sac tumor (YST) or non-gestational choriocarcinoma (CC)) (Figure 1) [2].

Ovarian yolk sac tumors (OYSTs) have the worst prognosis among MOGCTs [5]. Specific therapeutic strategies after the failure of platinum-based first-line and salvage regimens are needed. Risk factors for recurrence are stage (greater than I), age (older than 45 years) and management of treatment in non-referral centers [6].

Comprehensive knowledge of molecular biology might help to better understand the pathogenesis, the risk of relapse, and the development of potential innovative therapies [7]. However, only a few dedicated molecular investigation concern mOGCT.

Most therapeutic progress emerged from experience acquired in the treatment of testicular NSGCTs. However, some gender-specific molecular characteristics may be expected. In testicular GCTs, molecular structures amenable to targeted treatment approaches have been identified, and many of the available targeting agents have shown promising activity in vitro [8]. However, until today, results of preclinical models have rarely been translated into a benefit in clinical practice [9]. 

A national network for the rare ovarian tumors was created in 2011, supported by the French National Cancer Institute (INCa). This network provides diagnostic expertise and aims to improve the care of patients with these rare tumors using referral multidisciplinary tumor boards. It also facilitates recruitment for trials dedicated to only rare cancers with international effort [10,11].

In this context, in collaboration with EURACAN and EORTC SPECTA, we developed a project to better understand the molecular landscape of rare cancers. Here we focus on patients with OYSTs, who had been sequenced during the pilot phase of this project using retrospective samples; considering relapsed and disease-free patients.

## 2. Results

### 2.1. Population

A total of 10 patients with a yolk sac tumor diagnosed between 2007 and 2017 were selected, and their characteristics are presented in Table 1. Five patients (50%) were FIGO stage Ia, two (20%) stage IIIc and three (30%) were stage IV. Median age at diagnosis was 28 years (range 17–52). Nine patients had an oophorectomy and one patient had a radical surgery (bilateral salpingo-oophorectomy, total abdominal hysterectomy, omentectomy and lymphadenectomy) following a misdiagnosis initially identifying an ovarian high-grade serous carcinoma; this patient received a first cycle with carboplatin and paclitaxel. Nine out of ten received adjuvant bleomycin, etoposide, cisplatin (BEP)-based chemotherapy. Four patients (57%) planned to have four cycles of chemotherapy had their last cycle without bleomycin. Three patients relapsed, one of them was cured with a first-line salvage regimen, one died two years after the diagnosis after many lines of chemotherapy, and one progressed after first-line treatment and died (Table 2 and Table 3).

### 2.2. Molecular Characteristics and Abnormalities

Clinically significant alterations identified are presented in Table 4. Molecular analysis failed for one sample (patient #5). A gene alteration was identified in four out of nine profiled patients’ tumors. Three patients (33.3% of all patients) had potentially targetable activating mutations in *KRAS*, *KIT* and *ARID1A*. Patient #4 had seven concomitant mutations (KRAS G12V, *CCND2*, *FGF23*, *FGF6*, and *KDM5A* amplification, KIT D816A, ARID1A Q538). All 9 tested patients were MSI stable/low. Low tumor mutational burden (TMB) was identified for all of them, reporting a median of 3 mutations per megabase (Mut/MB) (range 0–5 Mut/MB). According to the FMI reports, treatment based on TMB score is only recommended for patients with 10 Mut/MB or more. Patient #7 had a *CRKL* amplification. Patient #10 had a KRAS D33E mutation.

Two of three patients with relapse had an oncogenic targetable mutation (*ARID1A* and *KRAS*). Among disease-free patients after first-line treatment, two out of seven harbored molecular alterations with one being targetable.

## 3. Discussion

This retrospective study reports results from 10 patients with YST diagnosed between 2007 and 2017. A complete response was observed for nine patients (90%) after first-line treatment. However, three patients relapsed (30%), one reported a complete response after HDCT, one patient died two years after diagnosis and one died one month after first line treatment.

Germ cell tumors are remarkably chemosensitive, and despite the high cure rates with initial and salvage chemotherapy, there remains a cohort of germ cell tumor patients who will eventually succumb to their progressive malignancy. New therapeutic agents are still needed for this patient population.

We report the molecular characteristics of nine YST patients with successful molecular analysis leading to potentially targetable oncogenic mutations identified for three patients (33.3% of all patients) in *KRAS*, *KIT* and *ARID1A*. Actionability is defined as a molecular alteration for which there is clinical or preclinical evidence of a predictive benefit from a specific therapy (in any cancer type). Interestingly, we were able to identify clinically relevant molecular alterations for all relapsed patients (patient #9 and #10). 

The main limitations of this study are the reduced sample size of our series and failure of molecular analysis for the patient with the poorest prognosis (patient #5). Dedicated studies are needed to confirm these results and to study benefit from specific therapeutic strategies after the failure of platinum-based first-line and salvage regimens. Additionally, further research should be done to tackle the correlation between the presence of a molecular alteration and the patients’ outcome.

Among mOGCTs, mutations in KIT have been previously identified exclusively in dysgerminoma (DG) at frequencies of 27% (6 of 22) and 24% (4 of 17), whereas no mutations have been reported in tumors of patients with pure or mixed histologies of IT and YST [14,15]. The success of KIT-targeting imatinib mesylate in the treatment of GIST [16] and chronic myelogenous leukemia [17] may be relevant for DG treatment [18]. However, a small trial investigating imatinib mesylate in patients with KIT-positive metastatic TGCT (*n* = 6) did not result in remission for any patient [19]. This phase II study defined KIT positivity by IHC as >10% of stained cells for KIT and the type of mutation was not defined. All previously reported mutations in DG were located in *KIT* exon 17 (D816V, D816H, and D816Y). Mutations in KIT activation loop (A-loop), including D816, have been reported to confer preclinical and clinical resistance to imatinib and sunitinib in GIST [20,21]. KIT exon 17 mutations, including D816, were reported to be sensitive to avapritinib [12]. 

In our cohort, patient #four had a KIT D816A mutation, but also a KRAS G12V hotspot mutation with amplification, *CCND2*, *FGF23*, *FGF6*, and *KDM5A* amplification, and ARID1A Q538* stop mutation. Among these mutations, three are actionable alterations (KIT D816A, KRAS G12V and ARID1A Q538*) with targeted therapies available. *KRAS* encodes a member of the RAS family of small GTPases and activating mutations in *RAS* can cause uncontrolled cell proliferation and tumor formation [22]. Preclinical evidence suggests that *KRAS* activation may predict sensitivity to MEK inhibitors, such as trametinib and cobimetinib tested in colorectal cancer [23]. 

*ARID1A* encodes a subunit of several different SWI/SNF protein complexes. SWI/SNF complexes regulate gene activity by a process known as chromatin remodeling. ARID1A is also recruited to DNA double-strand breaks (DSB) via its interaction with the upstream DNA damage checkpoint kinase ATR [24]. Tumors with this mutation may exhibit therapeutic vulnerability to PARP inhibitors according to preclinical results [13].

The other alterations are involved in cell growth and no targeted therapies directly address these genomic alterations. FGF6 and FGF23 encode a member of the fibroblast growth factor protein family with a role in muscle tissue regeneration and phosphate homeostasis, respectively. *CCND2* encodes Cyclin D2, which contributes to the regulation of the cell cycle G1-S transition. *KDM5A* encodes a lysine-specific histone demethylase that potentiates the expression of genes involved in cellular proliferation, senescence, angiogenesis, and migration.

All analyzed patients were microsatellite stable (MSS) and had a low TMB (median 3 Mut/MB, range 0–5 Mut/MB). The role of immune checkpoint inhibitors should be analyzed in these rare tumors. However, preliminary results from two studies investigating pembrolizumab and avelumab in male GCTs did not show significant efficacy [25,26]. Low immune infiltrate is associated with a poorer patient outcome in TGCT and improved knowledge in the tumor microenvironment may help for the development of immunotherapeutic strategies [27,28].

The YSTs often display complex and varied histological appearance, making differential diagnosis difficult, in particular between YST and endometroid or clear-cell carcinoma of the ovary. In our cohort, one patient has an initial misdiagnosis identifying high-grade serous carcinoma. A central pathology review confirmed the diagnosis of YST. Diagnosis of rare cancer is critical and may be challenging, and expert pathological review is recommended [11,29]. 

## 4. Materials and Methods 

### 4.1. Eligibility Criteria and Data Collection

In collaboration with EURACAN and EORTC SPECTA, data and samples were collected as per hospital policy. Patients signed a specific hospital consent to allow the use of their samples for future research, including genomic analysis, according to The International Council for Harmonization of Technical Requirements for Pharmaceuticals for Human Use (ICH) and applicable national laws. Two hospitals participated in the pilot phase of the Arcagen project: patients were seen at Centre Léon Berard (CLB), Lyon, France and Institut Bergonie, Bordeaux, France. We retrospectively identified a series of 10 patients with a yolk sac tumor (YST) in our database and reviewed their clinical, radiological, histological, and molecular characteristics. Selection criteria included good quality tumor blocs at initial diagnosis, diagnosis before 2017 and signed informed consent available for research. All available patients were analyzed and no additional selecting criteria were used.

### 4.2. Pathology Review

The ESMO guidelines were used as a standard for diagnosis to ensure consistency. All cases were examined by an expert pathologist in the diagnosis of YST as per ESMO guidelines through the national network for the rare ovarian tumors. 

### 4.3. Sample Workflow and Molecular Analyses

Samples were sent to the Foundation Medicine lab (Penzberg, Germany). The molecular analysis was performed using FoundationOne CDx. A single DNA extraction method from routine FFPE biopsy or surgical resection specimens, and 50–1000 ng underwent whole-genome shotgun library construction and hybridization-based capture. Using an Illumina^®^ HiSeq platform, hybrid capture–selected libraries were sequenced to high uniform depth. For each patient, the following variables are described in the molecular report provided by FMI: alteration type (SNV, deletion, amplification, fusion), alteration name, actionable gene alteration, therapies approved in EU (patient tumor type and other tumor types), tumor mutational burden (TMB) and microsatellite instability (MSI) status.

Actionability was defined as a molecular alteration for which there is clinical or preclinical evidence of a predictive benefit from a specific therapy (in any cancer type).

### 4.4. Regulatory Aspects 

Patients signed a specific hospital consent to allow the use of their samples for future research, including genomic analysis, according to The International Council for Harmonisation of Technical Requirements for Pharmaceuticals for Human Use (ICH) and applicable national laws. Data collections were in accordance with Act N°78–17 of 6 January 1978 on Data Processing, Data Files and Individual Liberties.

## 5. Conclusions

This study is the first part of the new research program (prospective SPECTA), exploring rare gynecological tumor molecular alterations and potential therapies. It highlights the feasibility to perform such molecular analysis in the routine for rare cancer settings but these results are very preliminary with a low sample size, and molecular analysis failed for one out of three patients who died.

Due to the lack of molecular evidence in rare tumors, other programs are needed with clinical trials and studies dedicated to better understanding rare gynecological tumor molecular profile and pathogenesis. Future analysis should also include RNA sequencing and phosphoprotein assessments.

## Figures and Tables

**Figure 1 cancers-13-00220-f001:**
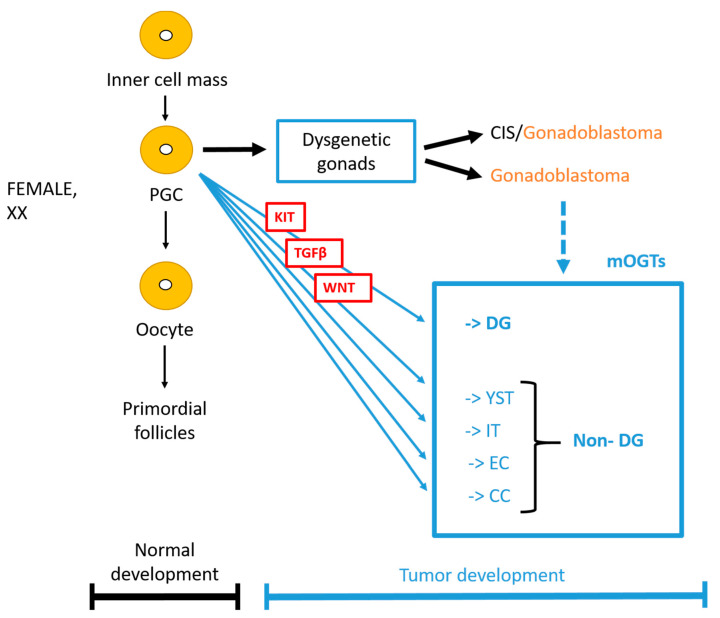
Illustration of germ cell tumor (GCT) development. CC: choriocarcinoma; CIS: carcinoma in situ; DG: dysgerminoma; EC: embryonal carcinoma; IT: immature teratoma; mOGCT: malignant ovarian GCT; PGC: primordial germ cell; YST: yolk sac tumor.

**Table 1 cancers-13-00220-t001:** Patient characteristics.

Patient Number	Age at Diagnosis (Year)	Medical History	Histology	Stage FIGO at Diagnosis
1	37 y	None	YST	Ia
2	37 y	None	YST	IVb
3	25 y	None	YST	IIIc
4	27 y	None	YST	IIIc
5	52 y	None	YST, first misdiagnosed as ovarian high grade serous carcinoma	IVb
6	29 y	None	YST	Ia
7	20 y	None	YST	IVa
8	35 y	Polycystic ovary syndrome	YST + mucinous cystadenoma	Ia
9	17 y	None	YST	Ia
10	26 y	Pure gonadal dysgenesis XY	YST + gonadoblastoma	Ia

Description of patient characteristics at diagnosis with FIGO stage and initial localization. YST: Yolk sac tumor.

**Table 2 cancers-13-00220-t002:** Patient initial treatment.

Patient Number	Surgery (Oophorectomy (A) vs. Radical Surgery (B))	Adjuvant Chemotherapy	Response to First Line Treatment	Relapse
1	A	4 cycles of BEP	Complete response	No
2	A	4 BEP	Complete response	No
3	A	3 BEP, 1 EP	Complete response	No
4	A	3 BEP, 1 EP	Complete response	No
5	B	1 cycle of Carboplatin paclitaxel then 3 cures of BEP	Progression	Died 1 month after the end of chemotherapy
6	A	3 BEP	Complete response	No
7	A	4 BEP	Complete response	No
8	A	3 BEP, 1 EP	Complete response	No
9	A	3 BEP 1 EP	Complete response	Relapsed after 5 year: aFP elevation and abdominal pain (one peritoneal node).
10	Bilateral oophorectomy	No	Complete response	Elevation of aFP 6 months after first surgery. Lesion of the upper bowel and iliac lymph node. Died 2 years after diagnosis

Description of initial treatment including type of surgery, chemotherapy, and initial response to treatment. BEP: Bleomycin, Etoposide, Cisplatin. EP: Etoposide, Cisplatin Radical surgery (B) involves bilateral salpingo-oophorectomy, total abdominal hysterectomy, omentectomy and lymphadenectomy.

**Table 3 cancers-13-00220-t003:** Treatment at relapse.

Patient Number	Treatment after First Relapse	Response to Treatment	Other Treatment	Last News
9	2 ICE with autologous stem cell injection	Complete response	No	Complete response (12/2019)
10	3 BEP, 1 EP	Progression with peritoneal carcinosis and spleen metastasis before surgery of residual active mass	3 VeIP docetaxel/gemcitabine (4 cycles) adriamycin/cyclophosphamid/avastin (2 cycles) carboplatin/ paclitaxel (1 cycle) endoxan/affinitor (1 cycle)	Died 2 year after diagnosis

Description of treatment at relapse and outcomes. BEP: Bleomycin, Etoposide, Cisplatin. ICE: Ifosfamide, Carboplatin, Etoposide. VeIP: Vinblastine, Ifosfamide, Cisplatin.

**Table 4 cancers-13-00220-t004:** Molecular characteristics.

Patient ID	Alteration Type	Alteration: Gene	MSI Status	TMB Status	TMB Muts/MB	Actionable Alteration ^+^	Potential Therapies *	Potential Therapies °
1	None	None	Stable	Low	3	None	None	
2	None	None	Stable	Low	3	None	None	
3	None	None	Stable	Low	0	None	None	
4	SNV + amplification	KRAS G12V amplification	Stable	Low	0	KRAS G12V amplification	Binimetinib, CobimetinibTrametinib	
	amplification	*CCND2* amplification				None	None	
	amplification	*FGF23* amplification				None	None	
	amplification	*FGF6* amplification				None	None	
	amplification	*KDM5A* amplification				None	None	
	SNV	KIT D816A				KIT D816A	dasatinib, Imatinib, Nilotinib, Ponatinib, Sorafenib, Sunitinib, binimetinib, cobimetinib, trametinib	ripretinib, avapritinib
	SNV	ARID1A Q538 *				ARIDA1	None	olaparib
5	Failed	Failed	Failed	Failed	Failed	Failed	Failed	
6	None	None	stable	Low	5	None	None	
7	amplification	*CRKL* amplification	stable	Low	4	None	None	
8	None		Stable	Low	4	None	None	
9	rearrangement	*ARID1A* rearrangement exon 19	stable	Low	0	ARID1A	None	olaparib
10	SNV	KRAS D33E	Stable	Low	4	KRAS D33E	binimetinib, cobimetinib, trametinib	

Description of clinically significant variants for each patient based on the FMI report. TMB: tumor mutational burden. ***** Specific therapies targeting this molecular alteration in other cancer types. **^+^** molecular alteration for which clinical or preclinical evidence of predictive benefit from a specific therapy (in any cancer type) from FMI report. ° Potential therapy not included in FMI report [12,13].

## Data Availability

The data presented in this study are available within the article in Table 1, Table 2, Table 3 and Table 4. No other data were created or analyzed.

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
