# Peer review of "Molecular Characterization of Ovarian Yolk Sac Tumor (OYST)"

_cancers, 2021, doi:10.3390/cancers13020220_

Round 1

Reviewer 1 Report

I am very grateful for the opportunity to review this manuscript. This study is the first molecular analysis of a yolk sac tumor was performed using FoundationOne CDx. It found that the yolk sac tumor is associated with a variety of genetic mutations. This provides useful information for future therapeutic strategies, especially for advanced and recurrent cases. However, it is also clear that BEP therapy, which has been used for some time, is still a good treatment option.

The questions for the authors are as follows.

How should you differentiate between BEP therapy and potential target therapy for patients with druggable target genes?

Although this study is rare cancers, it is still difficult to discuss many aspects of the study based on the results of the analysis of 10 cases.

Also, the inability to perform molecular analysis in case 5, which had the poorest prognosis, was the biggest drawback of this study.

Author Response

Dear reviewer,

Thank you very much for your comments.

Please find responses to your questions below:

  1. Druggable target genes is an opportunity for patients failing first line therapy and HDCT
  2. This study is exploratory and aims to assess the presence of targetable alterations in purpose to search them in clinical routine and to propose potential therapies after standard therapy failure. We agree that the sample size is low; however, this is due to the fact that this patient population is extremely rare. This study highlight the feasibility to perform such molecular analysis in rare cancer settings, even in routine care. We are also hoping that the prospective cohort will validate those exploratory results.

Reviewer 2 Report

This is an important report in that it identifies molecular alterations in a rare tumor type. The study is appropriately designed, interpreted and presented.  The authors were careful in their wording to assure that the reader understands that this a preliminary result and should not be over-interpreted.  The preliminary nature of this study is completely acceptable given that a rare tumor is being studied.  The inclusion of the patient diagnosis, treatment and outcomes add significance to the study. I only have two minor concerns:

1.  In the results section, the authors state that they selected 10 patients, however the Methods section does not describe the criteria used for the selection other than the patients were diagnosed with a yolk sac tumor. The authors should state whether these 10 patients represent all that were available or if they utilized additional criteria in selecting a subset of 10 patients out of the total.

2.  On line 123 in the discussion the authors state that they describe the molecular characteristics of 10 patients YST cohort, however since only 9 of the specimens were evaluable for molecular analysis, this number should be changed to 9 instead of 10 in this sentence.

Author Response

Dear reviewer,

Thank you very much for your comments.

Please find responses to your questions below:

  1. No additional criteria was used for patient selection, all available data was used. Based on your comment we added a statement to precise this point.

  2. You are right, molecular analysis failed for 1 patient and we adapted that sentence as suggested.

Please, feel free to share more comments with us to improve this study.

Reviewer 3 Report

The retrospective study entitled “Molecular characterization of Ovarian Yolk Sac Tumor (OYST)” presented 10 molecular analyses for 10 patients with OYST. The reviewer has a few comments which the authors may address:

1. The number of patients is too small. Each patient harbors different gene alterations. Random errors can be reduced by averaging over many observations.

2. The authors are suggested to discuss the microsatellite and tumor mutational burden in detail.

3. Does Table 4 represent the patients before or after treatment? Are these molecular alterations correlated with radiotherapy or chemotherapy?

4. Some grammatical errors and typos are present in the manuscript that needs correction. The authors are advised to proofread the paper by a native English speaker and modify the errors.

Author Response

Dear reviewer,

Thank you very much for your comments.

Please find responses to your questions below:

The retrospective study entitled “Molecular characterization of Ovarian Yolk Sac Tumor (OYST)” presented 10 molecular analyses for 10 patients with OYST. The reviewer has a few comments which the authors may address:

1. The number of patients is too small. Each patient harbors different gene alterations. Random errors can be reduced by averaging over many observations.

We are aware of the limited sample size for this study. This study is indeed exploratory in nature and results would need to be confirmed in the prospective phase that is currently recruiting.

2. The authors are suggested to discuss the microsatellite and tumor mutational burden in detail.

Thank you for your comment, it has been taken into account. Regarding MSI, FMI reports contain only information in binary form: stable/unstable. All our patients were MSI stable and, unfortunately, we cannot provide further information about MSI. However, we adapted the text to include TMB values as well: median value and range. Also, we explained for which TMB score FMI recommends the treatment.  

3. Does Table 4 represent the patients before or after treatment? Are these molecular alterations correlated with radiotherapy or chemotherapy?

Molecular characterisation is done from initial diagnosis sample and is not linked either to chemotherapy or radiotherapy

4. Some grammatical errors and typos are present in the manuscript that needs correction. The authors are advised to proofread the paper by a native English speaker and modify the errors.

Thank you for your comment. We tried to improve the quality of the language in the manuscript.

Please, feel free to share more comments with us to improve this study.

Round 2

Reviewer 3 Report

The authors revised the manuscript and responded to all comments from the reviewer.

Author Response

Thanks a lot for all your comments and your help